# Production of Abzymes in Th, CBA, and C57BL/6 Mice before and after MOG Treatment: Comparing Changes in Cell Differentiation and Proliferation

**DOI:** 10.3390/biom10010053

**Published:** 2019-12-28

**Authors:** Kseniya S. Aulova, Andrey E. Urusov, Ludmila B. Toporkova, Sergey E. Sedykh, Yuliya A. Shevchenko, Valery P. Tereshchenko, Sergei V. Sennikov, Thomas Budde, Sven G. Meuth, Nelly A. Popova, Irina A. Orlovskaya, Georgy A. Nevinsky

**Affiliations:** 1Institute of Chemical Biology and Fundamental Medicine, Siberian Branch of the Russian Academy of Sciences, 630090 Novosibirsk, Russia; amaya.rain.nsu@gmail.com (K.S.A.); urusow.andrew@yandex.ru (A.E.U.); sirozha@gmail.com (S.E.S.); 2Institute of Clinical Immunology, Siberian Branch of the Russian Academy of Sciences, 630090 Novosibirsk, Russia; toporkova12@mail.ru (L.B.T.); shevcen@ngs.ru (Y.A.S.); tervp@ngs.ru (V.P.T.); sennikov_sv@mail.ru (S.V.S.); irorl@mail.ru (I.A.O.); 3Westfälische Wilhelms-Universität, Institut für Physiologie I, Robert-Koch-Str. 27a, D-48149 Münster, Germany; tbudde@uni-muenster.de; 4Department of Neurology, Westfälische Wilhelms-Universität, Albert-Schweitzer-Campus 1, D-48149 Münster, Germany; sven.meuth@ukmuenster.de; 5Institute Cytology and Genetics, Siberian Branch of the Russian Academy of Sciences, 630090 Novosibirsk, Russia; nelly@bionet.nsc.ru; 6Novosibirsk State University, 630090 Novosibirsk, Russia

**Keywords:** Th mice, C57BL/6 mice, CBA mice, abzymes, immunization with MOG, hematopoietic progenitor differentiation, colony formation, catalytic antibodies

## Abstract

Till yet there is no data concerning mechanisms of autoimmune diseases development. Experimental autoimmune encephalomyelitis (EAE) prone C57BL/6 (T- and B-lymphocyte response), non-autoimmune CBA, and Th mice with T cell response were immunized with myelin oligodendrocyte glycoprotein (MOG_35–55_) to compare different characteristics of autoimmune reaction development. Bone marrow differentiation profiles of hematopoietic stem cells (HSCs), lymphocyte proliferation in various organs associated with the production of antibodies against DNA, myelin basic protein (MBP), and MOG, as well as abzymes hydrolyzing these antigens, were analyzed before and after immunization. Profiles of HSC differentiation [BFU-E (erythroid burst-forming unit (early erythroid colonies), CFU-E (erythroid burst-forming unit (late erythroid colonies), CFU-GM (granulocytic-macrophagic colony-forming unit), and CFU-GEMM granulocytic-erythroid-megakaryocytic-macrophagic colony-forming unit] and patterns of lymphocyte proliferation in different organs (brain, spleen, thymus, and lymph nodes) were very different for C57BL/6, CBA, and Th mice. We conclude that only C57BL/6 mice were predisposed to spontaneous and MOG-induced acceleration of EAE development. CBA mice are not prone to the development of autoimmune reactions. After immunization, Th mice demonstrate changes in several parameters similar to C57BL/6 and other to CBA mice; Th mice are more prone to developing autoimmune reactions than CBA mice. Our data may be important for understanding the combined presence in mice lymphocytes with T and B cell responses for spontaneous and induced autoimmune diseases.

## 1. Introduction

Multiple sclerosis (MS) is pathology of the central nervous system (CNS) defined by inflammatory and demyelinating processes associated with the occurrence of large numbers of macrophages and T lymphocytes. Numerous studies support the role of autoimmune (AI) mechanisms in the destruction of myelin, while the precise cause of MS remains unknown [1]. Some data show that activated myelin-reactive CD4^+^–T lymphocytes may be mediators of MS [1]. Several recent studies also validate the important role of B cells and autoantibodies (auto-Abs) against myelin autoantigens in MS pathogenesis [1,2,3].

The appearance of oligoclonal immunoglobulin G (IgG), increased amounts of antibodies (Abs), and the accumulation of B clonal cells in the cerebrospinal fluid (CSF), as well as the typical lesions in MS patients, provide key evidence for demyelination and the involvement of a humoral response [4]. Evidence from recent clinical studies in animal models demonstrates that auto-Abs against myelin components may be involved in antibody-mediated demyelination [3]. Auto-Abs against cell protein-oligodendrocyte progenitors may interfere with remyelination by removing or impeding these cells [5]. Both processes seem to play a crucial role in MS immunopathogenesis. In addition to MS-specific mechanisms, there may be common pathways in the development of different autoimmune diseases (AIDs) [6,7,8,9,10,11]. For example, the lymphocyte apoptosis level is usually increased in all patients with AIDs [6,7,8,9,10,11].

Artificial antibodies-abzymes against chemically stable analogues of transition states of different chemical reactions have been described [6,12,13]. Auto-Abs against several proteins, peptides, DNA, RNA, oligosaccharides, and various other components exist in healthy human blood and animals; their titers vary significantly, but these auto-Abs are catalytically inactive [6,7,8,9,10,11].

During the last 30 years, different auto-Abs with enzymatic activities have been identified. The production of auto-abzymes is a specific feature seen in patients with AIDs [6,7,8,9,10,11]. Similar to artificial abzymes, natural abzymes are Abs against enzyme substrates acting as protein haptens and mimicking transition states of chemical reactions. Anti-idiotypic abzymes against catalytic enzyme centers are observed in various catalytic activities [6,7,8,9,10,11]. Abzymes showing several different enzymatic activities are related to the earliest stages of AIDs. They are the most significant markers of disease onset and development for various AIDs in humans and mammals [6,7,8,9,10,11,14,15,16,17,18,19,20,21,22,23,24,25,26,27,28,29] including systemic lupus erythematosus (SLE) [24,25,26], MS [19,20,21,22,23,27,28,29], and experimental autoimmune encephalomyelitis (EAE) [17,18] in experimental mice. Enzymatic activities of abzymes are detectable at the pre-disease stage before the appearance of obvious AID markers and changes in proteinuria [6,7,8,9,10,11]. Auto-Abs titters for different anti-antigens in the pre-disease stage and at the onset of different AIDs usually correspond to the range of indices typical for healthy humans and experimental mice. The level of abzyme concentrations may indicate the pre-disease stage or disease onset, during which the increase in catalytic activities is connected to the development of deep AIDs symptoms [1,2,3,4,5,6,7,8,9,10,11,14,15,16,17,18]. To understand the disease development, it is important to identify all possible parallel and complementary mechanisms of AIDs development.

Current data display the appearance of myelin basic protein (MBP) and DNA-hydrolyzing abzymes in the blood of patients with MS [19,20,21,22,23], SLE [24,25,26], and several other AI pathologies [6,7,8,9,10,11]. Levels of IgGs from the CSF of MS patients hydrolyzing MBP, DNA, and polysaccharides are on average ~40–60 times higher than those taken from the sera of the same patients [27,28,29]. In AIDs, abzymes hydrolyzing MBP are dangerous since they can attack this protein from the axon’s myelin-proteolipid sheath and therefore play a harmful role in the pathogenesis of MS, SLE, and probably other diseases [6,7,8,9,10,11]. Abzymes with DNase activity are also harmful: they are cytotoxic and can penetrate the nucleus of the cell and hydrolyze nuclear DNA. This leads to cell death due to cell apoptosis, which, in turn, stimulates the development of AIDs [30,31,32]. Anti-DNA Abs and abzymes underlying AIDs are usually directed against histone-DNA nucleosomal complexes that appear during apoptosis, resulting in internucleosomal cleavage [33]. Similar to SLE pathology, high-affinity anti-DNA Abs were recently revealed as a major component of intrathecal Abs in the brains and CSF cells of MS patients [34].

It is well-established that all autoimmune pathologies arise after a self-tolerance breakdown (central or peripheral) and progress via multiple mechanisms. Additionally, autoimmune diseases have been suggested to originate from hematopoietic stem cell (HSC) defects [35]. It has been shown that the spontaneous and antigen-induced development of profound EAE pathology in C57BL/6 mice [17,18] and SLE pathology in MRL-lpr-lpr mice [14,15,16] is coupled with an immune system-specific reorganization. This reorganization includes irregular changes in the differentiation profile of bone marrow HSCs in combination with the production of abzymes hydrolyzing DNA, ATP, polysaccharides, and proteins. This phenomenon is one of the multiple mechanisms underlying self-tolerance breakdown.

We have identified several different EAE models, including C57BL/6 mice, which can be applied to mimic a specific facet of MS (for a review see [36,37,38]). In C57BL/6 mice, the disease takes a chronic-progressive clinical course and this model is characterized by specific T and B cell answers to antigens [37]. C57BL/6 mice were used recently to analyze possible mechanisms of spontaneous, myelin oligodendrocyte glycoprotein (MOG)- and DNA-accelerated EAE development [17,18]. Immunizing mice with MOG and DNA led to an accelerated EAE development, linked to various changes in differentiation profiles of HSC, lymphocyte proliferation, and apoptosis in organs associated with the production of abzymes hydrolyzing MBP, MOG, and DNA.

Here we used a model of spontaneous CNS autoimmunity which was generated by crossing myelin-specific T-cell receptor (TCR) transgenic mice and myelin-specific Ig heavy chain knock-in mice [39]. The offspring of those mice (Th mice) are characterized by spontaneous development of a severe form of EAE.

As shown in [14,15,16] in the study of SLE-prone MRL-lpr-lpr mice, in control CBA mice, in contrast to MRL-lpr-lpr mice, autoimmune reactions and significant changes in the differentiation profile after their immunization with DNA do not occur. At the same time, it was shown in [17,18] that in C57BL/6 mice with T- and B-cells response predisposed to EAE, the development of pathology is associated with a change in the profile of differentiation of bone marrow stem cells and the production of abzymes. Th mice with T-cells response also predisposed to spontaneous development of EAE [39]. Therefore, mice predisposed not predisposed to autoimmune to AIDS were selected as two control groups.

Taking all the data into account, we here for the first time carried out an extended analysis of spontaneous and MOG-induced changes in the differentiation profile of bone marrow HSC, lymphocyte proliferation in different organs, relative titers of Abs against MBP, MOG, and DNA, and relative activities of abzymes hydrolyzing these substrates for Th and CBA mice. These parameters were compared between EAE-prone C57BL/6 mice, non-autoimmune CBA mice, and Th mice with T cell response. We found significant differences in parameter changes between C57BL/6, CBA, and Th mice.

## 2. Material and Methods

### 2.1. Reagents

Different chemicals including proteins, bovine polymeric DNA, the Superdex 200 HR 10/30 column, and Protein G-Sepharose were obtained from Sigma-Aldrich (Munich, Germany) and GE Healthcare (New York, NY, USA). Purified human MBP containing 18.5 kDa from RCMDT (Moscow, Russia) and MOG_35−55_ from EZBiolab (Berlin, Germany) were used. These preparations were free from oligosaccharides, lipids, nucleic acids, and other possible contaminants.

### 2.2. Experimental Animals

Th, C57BL/6 inbred, and CBA mice (3 months of age) were housed in the mouse breeding facility at the Institute of Cytology and Genetics (ICG) in standard pathogen-free conditions (including a special system for protection against bacterial and viral infections). All animal experiments were carried out according to protocols of the Bioethical Committee of the ICG in agreement with recommendations of the European Committee for the humane principles of work with animals (European Communities Council Directive 86/609/CEE). The Bioethical Committee of the ICG approved our study according to the European Communities Council Directive 86/609 guidelines.

### 2.3. Immunization of Mice

Mice were immunized with MOG according to previously published protocols [17,18,38]. At time zero (day 1), mice were immunized by subcutaneously injecting MOG (10 μg per mouse) in the cervical region using 40 μL of Freund’s complete adjuvant containing pertussis toxin (400 ng/mouse; *Mycobacterium tuberculosis*). On day 2, an additional 20 μL of pertussis toxin (400 ng/mouse) were injected in a similar manner. The weight of mice and levels of proteinuria (protein concentration in the urine, mg/mL) were analyzed as described before [14,15,16,17,18]. Proteinuria was measured with the Bradford assay using bovine serum albumin as standard. For purification of Abs and other analyses, 0.7–1 mL of blood were collected after standard mice decapitation.

### 2.4. ELISA of Anti-Protein, Anti-DNA, and Antibodies

Anti-DNA (plasma was diluted 100-fold), anti-MBP and anti-MOG (for both plasma was diluted 50-fold), and Abs concentrations were analyzed using ELISA as described before [17,18]. After treating blood plasma with rabbit-specific anti-mouse Abs conjugated with horseradish peroxidase, the mixtures were incubated with H_2_O_2_ and tetramethylbenzidine. The reaction was stopped by adding of H_2_SO_4_ and the optical density (A_450_) of the solutions was estimated using a Uniskan II plate reader (MTX Lab Systems, New York, NY, USA).

The relative concentrations of Abs against MOG, MBP, and DNA, in the samples were estimated from differences in optical density (A_450_) between control and experimental samples. Controls with DNA, MOG, and MBP, but without serum samples and with no interaction between Abs and antigens, gave the same results.

### 2.5. IgG Purification

Electrophoretically homogeneous IgGs were purified by sequential affinity chromatography of plasma proteins on Protein G-Sepharose and FPLC gel filtration as described before [17,18,19,20,21,22,23,24,25]. Abs were filtered (Millex; 0.1 μm) into sterilized tubes to protect them from viral and bacterial contamination. The absence of bacterial and viral colonies in IgG preparations was verified as described before [28,29]. SDS-PAGE of IgGs in non-reducing conditions (0.1%) was performed using 4–15% gradient gels. Intact IgGs were visualized by silver staining as described before [25,26,27,28,29]. To exclude possible admixtures of canonical enzymes, catalytic activities of IgGs were analyzed after SDS-PAGE using extracts of the gel fragments according to previous experiments [17,18,19,20,21,22,23,24,25]. Catalytic activity in peaks after SDS-PAGE was found only in the protein bands of intact IgGs; no other protein band peaks of protease or DNase activities were revealed.

### 2.6. DNA-Hydrolyzing Activity Assay

DNase activity of Abs was analyzed as in previous experiments [19,20,21,23]. The reaction mixtures (20 μL) contained 20 mM Tris-HCl (pH 7.5), 20 mM NaCl, 5 mM MgCl_2_, 1 mM EDTA (ethylenediaminetetraacetic acid), 20 μg/mL supercoiled (sc) pBluescript, and 0.03–0.2 mg/mL of IgGs. The mixtures were incubated for 1–12 h at 37 °C. Products of DNA hydrolysis were analyzed using electrophoresis on 0.8% agarose gels. Photographs of ethidium bromide-stained gels were analyzed using Gel-Pro Analyzer v9.11. The relative catalytic activity was estimated from the difference in the percentages of intact supercoiled DNA (scDNA) and its relaxed form, considering a distribution of DNA between these bands after incubation of scDNA in the absence of Abs. All initial reaction rates were analyzed using the linear parts of the time dependencies (15–40% of DNA hydrolysis) and concentrations of IgGs for every preparation (20–40% of DNA hydrolysis). A complete conversation of scDNA to its nicked form was taken as 100% of the activity. Finally, the relative activities (% of the hydrolysis) were recalculated to the same standard time and Abs concentration.

### 2.7. Protease Activity Assay

The reaction mixtures (10–40 μL) contained 20 mM Tris-HCl (pH 7.5), 0.7–1.0 mg/mL of either MOG or MBP, and 0.01–0.2 mg/mL of IgGs as described before [17,18,19,20,21,22,23,24]. They were incubated for 1–24 h at 37 °C. The protein cleavage products were analyzed by SDS-PAGE using a 3–15% gradient or 12% gels and Coomassie R250 staining. The gels were scanned and then quantified using GelPro v3.1 software. The relative activities of different IgGs were calculated from a decrease in the percentage of initial protein transited to its different hydrolyzed forms; the hydrolysis of proteins incubated without IgGs was taken into account. All initial rates of protein hydrolysis were estimated using the pseudo-first-order reaction condition including the time course linear regions and concentrations of IgGs (20–40% hydrolysis of the proteins).

### 2.8. In Culture Analysis of Bone Marrow Progenitor Cells

Samples of bone marrow were obtained from mouse femurs and the ability of the bone marrow cells to form colonies was estimated as described previously [14,15,16,17,18]. Four dishes per mouse (2 × 10^4^ cells) were grown using the standard methylcellulose-based M3434 medium (StemCell Technologies, Canada) specific for mouse cells. The medium contained stem cell factor, erythropoietin (EPO), interleukins IL3−, and IL6−. The relative number of CFU-GM, CFU-GEMM, CFU-E, and BFU-E colonies was calculated in the samples after 14 days of incubation at 37 °C and 5% CO_2_ in a humidified incubator as described previously [14,15,16,17,18].

### 2.9. Evaluation of Lymphocytes in Samples of Different Mouse Tissues

In vitro analysis of the sum of B and T lymphocyte proliferation was carried out as described before [14,15,16,17,18]. Cells isolated from thymus, spleen, bone marrow, and lymph nodes were cultivated in 96-well flat-bottom plates (Trasadingen, Switzerland,) containing RPMI1−640 medium supplemented with 10 mM HEPES buffer, 10% fetal calf serum, 2 mM L-glutamine, 0.5 mM 2-mercaptoethanol, 100 μg/mL benzylpenicillin, and 80 μg/mL gentamicin. After a 64 h incubation period, a solution (15 μL) containing 5 mg/mL MTT (3-(4,5-dimethylthiazol^2−^-yl)^2−^,5-diphenyl tetrazolium bromide) was added to each well and incubated at 37 °C for an additional 4 h. Then, plates were centrifuged at 1200× *g* for 10 min and solutions were removed. Cells were precipitated by adding DMSO (200 μL); the mixtures were resuspended and incubated in darkness at 23 °C for 15 min. The relative cell amount was analyzed spectrophotometrically at 492 nm (A_492_).

### 2.10. Statistical Analysis

The values obtained are given as the mean ± S D of at least three to four independent experiments for each mouse, averaged over 7 different mice. Differences between the examined samples and the three mouse groups were analyzed using Student’s *t*-test; *p* ≤ 0.05 was considered as statistically significant.

## 3. Results

### 3.1. Choosing a Model for Studying the Mechanism of EAE Development

According to the literature, the T cell immune system plays a leading role in human MS pathogenesis, while the B cell system is also important for disease development [1]. B lymphocytes provide the humoral immunity components of the adaptive immune system by secreting Abs [40]. Unlike the other two classes of lymphocytes, namely T cells and natural killer cells, mature B cells in the bone marrow have membrane receptors that allow them to bind to a specific antigen against which they will initiate an antibody response. MOG-induced EAE in C57BL/6 mice with T and B cell response is frequently used as a model of human MS [36,37,38]. Studies show that immunizing C57BL/6 mice with MOG significantly changes the differentiation profiles of HSCs and the lymphocyte proliferation in different organs, and leads to the production of Abs against MBP, MOG, and DNA harmful for animals as well as abzymes efficiently hydrolyzing MBP, MOG, and DNA [17,18]. In contrast to the C57BL/6-line, Th mice are characterized with T cell responses to antigens [39]. CBA mice are not prone to developing AIDs. Consequently, we set out to compare changes in all of the above-mentioned parameters for EAE, CBA, and Th mice. We compared the development of EAE over time using previously obtained data from C57BL/6 mice and two new experimental groups: Th untreated control/Th MOG-treated mice and CBA untreated control/CBA MOG-treated mice. The same experiments were performed earlier and well-reproducible data on the analysis of all parameters for untreated control and MOG-treated C57BL/6 mice were available [17,18,41,42] for comparison with the new results.

### 3.2. Changes in Proteinuria and Weight of Mice

Changes in the weight of Th and CBA mice before and after MOG treatment were analyzed from the day of immunization (time zero, at three months of age) for 45–85 consecutive days (Figure 1A,B). We found that immunizing Th and CBA mice with MOG slows weight gain: by day 45 the weight of treated Th mice was ~1.2-fold less and the weight of treated CBA mice ~1.1-fold less than the weight of non-treated mice (Figure 1). Interestingly, immunizing C57BL/6 mice also led to a weight decrease over time compared to untreated animals, but to a much lesser extent (Figure 1A).

In different animal AI models, the development of pathologies usually correlates with an increase in the concentration of proteins in the urine (proteinuria) [14,15,16,17,18]. Untreated control non-autoimmune CBA mice were characterized by low levels of proteinuria at three months of age (1.6 ± 0.08 mg/mL) and levels did not change remarkably over the next 45 days (Figure 1C). Interestingly, C57BL/6 mice showed significantly higher proteinuria (up to 7.22 ± 0.9 mg/mL) at three months of age even before treatment with MOG [17,18,38,41,42]. Upon spontaneous development of EAE, levels of proteinuria almost did not change in untreated control C57BL/6 mice before day 20, and then gradually increased to 17.0 mg/mL by day 63. After treating C57BL/6 mice with MOG, urine protein levels increased 3.8-fold up to 277 ± 3.0 mg/mL (Figure 1C). Interestingly, until three months of age levels of proteinuria in Th mice were similar to levels seen in C57BL/6 mice (8.1 ± 2.9 mg/mL; Figure 1). Unlike CBA mice, Th mice demonstrated a small proteinuria increase of 1.3-fold during the first 73 days before immunization, and an increase of 1.7-fold after immunization (Figure 1C). In the case of non-autoimmune CBA and BALB mice, levels of proteinuria are low and remain almost unchanged for at least 12 months [14,15,16]. As we have shown in previous studies [14,15,16,17,18,41,42], a high level of proteinuria until three months of age and a proteinuria increase in the time before immunization may indicate that these mice are prone to spontaneous development of various AIDs. However, the increases in proteinuria for autoimmune prone C57BL/6 mice during spontaneous EAE development and after immunization with MOG were significantly higher than those seen in Th and CBA mice. Therefore, Th mice may be less prone to spontaneous autoimmune reactions than C57BL/6 mice.

### 3.3. Hematopoietic Progenitor Colony Formation

Spontaneous and MOG-induced development of EAE in C57BL/6 mice leads to significant changes in the differentiation profile of hematopoietic progenitors: BFU-E (erythroid burst-forming unit, early erythroid colonies), CFU-E (erythroid burst-forming unit, late erythroid colonies), CFU-GM (granulocytic-macrophagic colony-forming unit), CFU-GEMM (granulocytic, erythroid, myeloid colony-forming unit), and lymphocyte proliferation levels [17,18,41,42].

In untreated Th and CBA control mice, the average number of BFU-E colonies gradually decreased over time and was 2.5- and 1.6-fold lower, respectively, between days 56–63 compared to time zero (Figure 2A). In contrast, during spontaneous EAE development in C57BL/6 mice, the relative number of BFU-E colonies did not change between days 0–20 and then increased 2.1-fold until day 63 (Figure 2A). After immunizing CBA mice, the number of BFU-E colonies showed almost no change during the duration of the experiment (Figure 2B). After C57BL/6 mice were immunized with MOG, EAE onset corresponded to days 7–8, while the acute stage usually occurred between days 14–20 and the remission stage started between days 25–60 [36,37,38]. Treating Th and C57BL/6 mice with MOG led to changes in BFU-E colonies in opposite directions. Changes in the number of BFU-E colonies in Th mice were complex: similar to spontaneous EAE development in C57BL/6 mice the number of BFU-E colonies first decreased 1.3-fold until day seven, then increased 1.3–1.5-fold between days 14–20 corresponding to acute stages of EAE in C57BL/6 mice, followed by a great reduction until day 30 and a renewed increase of 1.9-fold until day 40 (Figure 2B). Changes in the number of BFU-E colonies of C57BL/6 mice have a reverse character, but are relatively simple: the number of BFU-E colonies decreased 1.9-fold at disease onset and in the acute phase of EAE (at 7–20 days), and then continuously increased 1.8-fold until day 63 compared to time zero of the experiment (Figure 2B). Thus, changes observed for the number of BFU-E colonies in the three strains of mice before and after their immunization with MOG differed substantially.

During spontaneous EAE development in CBA mice, the number of CFU-E colonies decreased 3.4-fold until day 62, while in Th mice they increased 1.7-fold (Figure 2C). Before MOG treatment, there was a 4.1-fold increase in the relative number of CFU-E colonies during the first 63 days in C57BL/6 mice (Figure 2C). Before treatment, Th and CBA mice demonstrated similar changes in CFU-E colonies until day 30, while Th mice showed an increase in the number of colonies similar to C57BL/6 mice thereafter.

Immunizing CBA and Th mice with MOG led to a weak change in the number of CFU-E cells; the increase in number until day 60 was only 1.5- and 1.9-fold, respectively (Figure 2D). In contrast, after immunizing C57BL/6 mice with MOG there was a 6.5-fold increase in the relative number of CFU-E colonies between days 6–10, with a slight 1.1-fold decrease from day 20 to day 63 (Figure 2D). Thus, CBA and Th mice show similarities and significant differences in CFU-E colony development after MOG immunization compared to C57BL/6 mice (Figure 2C,D).

In the period before MOG immunization, the relative number of CFU-GM colonies over time increased 1.8-fold for CBA mice and 1.4-fold for C57BL/6 mice, while there was a minimal change for Th mice (Figure 3A). After MOG treatment, CFU-GM cells decreased 1.6-fold in CBA mice, while their number strongly increased between days 7–20 in Th mice, followed by a remarkable decrease (Figure 3B). Interestingly, for C57BL/6 mice, the opposite effect was observed—a sharp 1.6-fold decrease in the number of colonies between days 7–10 and a subsequent increase between days 30–60 (Figure 3B).

Before MOG treatment, the number of CFU-GEMM colonies decreased in a comparable way for Th mice (1.3-fold) and C57BL/6 mice (2.3-fold) (Figure 3C), while they significantly increased for CBA mice up to days 20–30 (Figure 3C). Thus, before treatment, Th and C57BL/6 mice showed a similar development in the number of CFU-GEMM colonies. After immunization with MOG, these cells decreased in all three mouse strains (Figure 3D). However, for Th and CBA mice there was a gradual decrease in the relative number of CFU-GEMM cells over time, while a sharp decrease was observed for C57BL/6 mice between days 7–20, followed by an increase (Figure 3D).

For a more detailed comparison of the differences in bone marrow stem cell differentiation for the three lines of mice, the relative number of the four types of colonies and their total amount before and after immunization was estimated. For analysis, the total number of the four types of colonies was taken as 100% in all cases. Interestingly, considering the standard deviation for CBA mice, the total number of all cells before and after MOG treatment was almost the same (difference 1.05-fold; Figure 4A). In contrast, immunizing C57BL/6 mice (2.61-fold) and Th mice (1.59-fold) resulted in significant growth of the total number of blood precursors (Figure 4B,C). After MOG treatment, an average increase in the number of BFU-E cells occurred for all three mouse strains (Figure 4). However, for CBA mice the increase was significantly lower (1.1-fold) than that for C57BL/6 mice (1.8-fold) and Th mice (2.1-fold). After immunization, a remarkable increase in CFU-GM colonies was observed only for C57BL/6 mice (1.2-fold) and Th mice (1.4-fold), while CBA mice demonstrated a slight decrease (1.1-fold; Figure 4). The average number of CFU-GEMM colonies after immunization decreased for all three mouse strains: 1.1-fold for CBA mice, 1.6-fold for C57BL/6 mice, and 2.0-fold for Th mice. The main difference between the three lines of mice was a significant increase in the relative number of CFU-E cells after immunization for EAE prone C57BL/6 mice by a factor of 6.1, while the relative amount decreased for CBA mice (1.25-fold; Figure 4A) and Th mice (1.7-fold; Figure 4C).

Thus, very different changes in the differentiation profiles of the four types of hemopoietic cell precursors are observed for C57BL/6, CBA, and Th mice over time before and after MOG immunization. We cannot exclude that the development of autoimmune reactions to some extent depends on changes in the relative amount of all four types of hematopoietic blood precursors. For C57BL/6 mice, the sharp increase (6.1-fold) in the number of CFU-E cells may be the most important factor for the development of EAE. Nevertheless, an increase in specific lymphocyte fractions and their total number in the brain and other organs of mice is also may be very important for the development of autoimmune reactions and the production of auto-Abs and can be harmful to mice abzymes.

### 3.4. The Relative Content of Lymphocytes in Various Organs of Mice

Before immunization with MOG, spontaneous development of EAE in C57BL/6 mice led to a strong increase in the number of T and B lymphocytes in the brain (1.9-fold; Figure 5A) and spleen (1.8-fold; Figure 5C). However, in the brain, the average relative number of lymphocytes increased evenly over time, while in the spleen it increased only after day 20. After C57BL/6 mice were immunized with MOG, a small lymphocyte decrease (1.1-fold) was observed in the brain. Nevertheless, the curves for lymphocyte increase before and after immunization were very close (Figure 5B).

After immunizing Th mice, a significant lymphocyte decrease (1.9–2.4-fold) was observed between days 10–30 (Figure 5B). Non-autoimmune CBA mice demonstrated a similar decrease in brain lymphocytes before and after immunization (Figure 5A,B). It is possible that the lymphocyte decrease in the brains of Th and CBA mice, in contrast to the increase observed in C57BL/6 mice, leads to a reduction of specific lymphocytes producing auto-Abs and abzymes in their brain, blood, and other various organs.

The number of lymphocytes in the spleen of C57BL/6 mice after immunization increased significantly by day 14 (2.6-fold), followed by a noticeable decrease (Figure 5D). A slight increase in the average relative number of lymphocytes in the spleen of CBA mice was observed between days 7–14, followed by a remarkable decrease (Figure 5D).

Interestingly, lymphocyte numbers in the spleen of Th mice before and after immunization were almost the same (Figure 5C,D). We assume that the spleens of CBA and Th mice respond more poorly to the immunization with MOG compared to C57BL/6 mice.

Before treating C57BL/6 mice, we observed a relatively weak and even increase in the number of lymphocytes in the thymus (1.3-fold) and a lymphocyte decrease in the lymph nodes (1.8-fold) (Figure 6A,C). Before MOG immunization, Th and CBA mice showed a weak and nearly gradual increase in the number of lymphocytes in the thymus (Figure 6A) and a very slight change in their number in the lymph nodes (Figure 6C). After immunization, the relative percentage of lymphocytes in the thymus of CBA mice was almost unchanged over time (Figure 6B), while there was a noticeable increase in their number in the lymph nodes between days 7–14, followed by a strong decrease (Figure 6D).

For Th mice, the curve of the percentage of lymphocytes after immunization is similar to the curve before immunization, with a small increase peaking at about day 21 (Figure 6C,D). Interestingly, the relative number of lymphocytes in the lymph nodes of Th mice before and after immunization is nearly the same (Figure 6C,D).

Thus, the change in the relative number of lymphocytes in different organs of C57BL/6, CBA, and Th mice before and after immunization is different. However, CBA and Th mice demonstrate greater similarities for this parameter, with very different values compared to C57BL/6 mice.

### 3.5. The Relative Content of Abs Against Proteins and DNA

The blood sera of healthy humans and animals usually contain auto-Abs against DNA and different proteins in low concentration [7,8,9,10,11]. The relative concentrations of anti-DNA Abs of non-autoimmune BALB and CBA mice (at 3–10 months of age) as well as healthy MRL-lpr/lpr mice (at 2–3 months of age) are usually low and vary in the range from 0.03 to 0.04 A_450_ units [14,15,16]. The average concentration of anti-DNA Abs at three months of age in sera of C57BL/6 mice is ~0.11 A_450_ units, slowly rising approximately 1.4-fold for untreated mice (0.15 A_450_ units) and 4.5-fold for MOG immunized mice (0.5 A_450_ units) until day 63 (Figure 7A-a1). In non-autoimmune CBA mice the concentration of anti-DNA Abs in sera at three months of age was 3.7-fold lower than for C57BL/6 mice and changed very little over time (Figure 7A-a2). After immunizing CBA mice, the Abs concentration increased 2-fold until day 73 (Figure 7A-a2). It was somewhat unexpected that the concentration of Abs against DNA in Th mice was 14.7- and 4.3-fold lower at three months of age than the concentration in C57BL/6 and CBA mice, respectively (Figure 7A-a2). Until day 75 after immunization, the concentration of anti-DNA Abs in Th mice increased 3.1-fold.

Figure 7B shows time-dependent changes in Abs against MOG in the sera of untreated and treated C57BL/6, CBA, and Th mice. The average relative concentration of anti-MOG Abs in untreated spontaneously diseased C57BL/6 mice increased 5.7-fold during 63 days, while the treatment accelerated the increase of these Abs concentrations over a period of 6–10 days. At day 63 they had increased 9.1-fold compared to time zero (Figure 7B-b1).

Before immunizing CBA mice, no substantial change in anti-MOG Abs was observed, while treatment led to a 1.5-fold increase only (Figure 7B-b2). The anti-MOG Abs development for Th mice was similar (Figure 7B-b2).

The initial concentrations of anti-MBP Abs in the blood serum of C57BL/6, CBA, and Th mice at three months of age were comparable (~0.02 A_450_/_mL_). During spontaneous EAE development, anti-MBP Abs gradually increased 2-fold in C57BL/6 mice. Immunizing C57BL/6 mice with MOG led to a strong 11-fold increase (Figure 7C-c1). The concentration of anti-MBP Abs in sera of CBA and Th mice showed almost no change over time before immunization, while it was increased 2-fold after treatment (Figure 7C-c2). Thus, the titers of Abs against DNA, MBP, and MOG are significantly lower before immunization in CBA and Th mice at three months of age than those in C57BL/6 mice (Figure 7). Additionally, the significant titer increase before immunization was observed only for EAE prone C57BL/6 mice, while CBA and Th mice showed no noticeable changes. After immunizing C57BL/6 mice, the concentration of Abs against DNA, MBP, and MOG increased 4.5–11.0-fold, while for CBA and Th mice it only increased 1.5–2.0-fold. These results accentuate a predisposition for significant spontaneous development of autoimmune reactions only for C57BL/6 mice, an acceleration which occurs after MOG treatment.

### 3.6. Criteria Analysis of the Association Between Activities and Antibodies

As shown earlier, Abs from sera of C57BL/6 mice possess DNase, MOG-, and MBP-hydrolyzing activity. Here, were purified IgGs from individual CBA and Th mice by affinity chromatography of blood sera proteins on Protein G-Sepharose using conditions to remove nonspecifically bound proteins as described before [17,18]. Then, IgGs were additionally purified using an FPLC gel filtration. To analyze Abs homogeneity, IgGs from individual mice of each group and equal amounts of Abs from sera of Th mice (th-IgG_mix_) and CBA mice (cba-IgG_mix_) were used. The electrophoretic homogeneity of th-IgG_mix_ and cba-IgG_mix_ was shown by SDS-PAGE with silver staining (Figure 8A).

As demonstrated before, the method of isolation developed by us excludes contamination of IgGs with canonical enzymes [14,15,16,17,18,19,20,21,22,23,24,25,26,27,28,29]. Applying several very strict criteria that we developed previously [6,7,8,9,10,11,43], we were able to show that DNA-, MOG-, and MBP-hydrolyzing activities are own properties of IgG abzymes derived from sera of spontaneously diseased and MOG-treated C57BL/6 mice, and are not due to co-purified canonical DNases or proteases [17,18,41,42]. Here, we additionally prove that IgG_mix_ preparations, corresponding to the mixtures of IgGs of Th and CBA mice treated with MOG, do not contain impurities from canonical enzymes. To test the th-IgG_mix_ and cba-IgG_mix_, gel strips were cut into 2–3 mm wide fragments after Abs separation by SDS-PAGE. MOG-, histones-, and DNA-hydrolyzing activities were analyzed using extracts of proteins from the separated gel fragments (Figure 2C,D). DNA-, MOG-, and MBP-hydrolyzing activities were found only in the fragments of gels containing intact IgGs. SDS usually destroys any complex proteins, while the electrophoretic mobilities for canonical DNases and proteases (28–32 kDa) are significantly higher than for IgGs (~150 kDa). Thus, the detected DNA, MOG-, and MBP-hydrolyzing activities in gel fragments correspond only to intact IgGs (Figure 2B) provide direct evidence that IgGs in MOG-treated CBA and Th mice possess DNase and protease activities.

### 3.7. Time-Dependent Changes in IgGs Catalytic Activities

We estimated the changes over time in the average relative activities of IgGs corresponding to the CBA and Th experimental groups and compared them with our existing data from C57BL/6 mice [17,18,41,42]. As described above, the increased concentrations of abzymes with different enzymatic activities are a significant marker of disease onset and development for various AI reactions [6,7,8,9,10,11,14,15,16,17,18,19,20,21,22,23,24,25,26,27,28,29]. It is known, that after treating C57BL/6 mice with MOG the onset of EAE corresponds to days 7–8, the acute phase usually occurs between days 14–20, and the remission stage begins around 25–30 days [36,37,38]. As shown earlier [17,18,41,42], EAE prone C57BL/6 mice demonstrated a gradual and nearly linear 6.8-fold increase of DNase activity before immunization (Figure 9-a1). After treating C57BL/6 mice with MOG, this activity increased from day 7 onwards, and at 21 days it was 24.4-fold higher than at time zero, followed by a significant decrease between days 35–63 (Figure 9-a1).

At time zero, DNase activity of IgGs in Th mice was approximately 4-fold lower than that in C57BL/6 mice (Figure 9A-a1). Before immunizing Th mice, the change over time for DNase activity of IgGs was relatively weak (Figure 9A-a1) and similar to the change in the concentration of anti-DNA Abs (Figure 7A-a2). However, after treating Th mice with MOG DNase activity of IgGs increased from day 7 onwards and was 32-fold higher at days 20–25 than at the beginning of the experiment.

Non-autoimmune CBA mice demonstrated an absence of reliably tested DNase activity before immunization, and their treatment with MOG led to a slow, gradual activity increase over time (Figure 9A-a2). Interestingly, the increase of anti-DNA Abs concentration after immunization of CBA mice occurred mainly within 7–10 days, followed by a plateau (Figure 7A-a2). Therewith, the maximal DNase activity of IgGs in CBA mice is 6- and 19-fold lower than the activity seen in Th and C57BL/6 mice, respectively.

At three months of age, the concentration of anti-MOG Abs in C57BL/6 mice was 3.7- and 15.7-fold higher than in CBA and Th mice, respectively. Despite the very low concentration of Abs against MOG in the blood of Th mice at time zero, their IgGs showed a reliably tested MOG-hydrolyzing activity, which was 4.8-fold lower than in C57BL/6 mice (Figure 9B-b2). Before immunization, CBA mice did not demonstrate MOG-hydrolyzing activity (Figure 9B-b2). After MOG treatment, there was a gradual increase of MOG-hydrolyzing activity in Th and CBA mice, while the maximal activity seen in CBA mice was 2.9-fold lower than in Th mice (Figure 9B-b2). Interestingly, the maximal MOG-hydrolyzing activity in C57BL/6 mice was approximately 4.7- and 13.5-fold higher than in Th and CBA mice, respectively (Figure 9B-b2). It should be noted, that after immunization the curve characterizing the increase over time of relative anti-MOG Abs concentrations (Figure 7B) and the curve describing the enhanced MOG-hydrolyzing activity (Figure 9B) have a similar progression.

The time-dependent changes in the concentration of Abs against MOG and MBP in sera of CBA and Th mice have similar patterns, but they differ from those in C57BL/6 mice (Figure 7). Some similarities between CBA and Th mice were visible in the patterns of changes over time concerning the hydrolysis of MOG and MBP by Abs from the blood and were again very different from those in C57BL/6 mice (Figure 9). The highest concentrations of anti-MBP Abs in sera of CBA and Th mice were about 2-fold higher than for Abs against MOG (Figure 7). The relative efficiency of Abs hydrolyzing MOG (~4%) and MBP (~5%) in Th mice was comparable at the beginning of the experiment (Figure 9B,C). However, after immunizing Th mice, the efficiency of MOG hydrolysis increased 5.3-fold (21.3%), and the efficiency of MBP hydrolysis increased 2.3-fold (11.3%). After immunizing CBA mice, abzymes hydrolyzing MOG and MBP appeared in the blood, but their activities were 1.7–3.9-fold lower than in Th mice. Thus, the maximum activities of blood-derived abzymes hydrolyzing MOG and MBP were 4.7–8.8-fold lower in Th mice and 14.3-fold lower in CBA mice compared to C57BL/6 mice.

## 4. Discussion

It has been shown that various AI pathologies arise after self-tolerance breakdown (central or peripheral) via multiple mechanisms. EAE-prone C57BL/6 and SLE-prone MRL-lpr/lpr mice are predisposed to the spontaneous development of AI reactions. The spontaneous development of EAE in C57BL/6 mice and SLE in MRL-lpr/lpr mice, as well as the acceleration of AI reactions after treatment with MOG and DNA, respectively, is characterized by very similar changes in the differentiation of HSCs, which are associated with the onset of abzymes production [14,15,16,17,18,41,42]. Taking this into account, we have a strong interest to compare the parameters characterizing the development of AI reactions and the features of the production of abzymes in mice prone and not prone to developing AIDs, including non-autoimmune CBA mice and Th mice with T cell answer.

Interestingly, the differentiation profile of bone marrow stem cells in C57BL/6, CBA, and Th mice followed different patterns before and after their immunization with MOG (Figure 2; Figure 3). However, prior to MOG treatment, we observed a similar decrease over time for BFU-E (Figure 2A) and CFU-E cells (Figure 2B) in CBA and Th mice, while these cells significantly increased in C57BL/6 mice. The decrease in the relative number of these cells in CBA mice over time was somewhat compensated due to an increase in the relative number of CFU-GM and CFU-GEMM cells (Figure 3A,C). Moreover, the total number of the four types of precursor cells in the brain before and after MOG treatment was almost the same (difference 1.05-fold; Figure 4A). Before immunization, C57BL/6 and Th mice exhibited approximately the same decrease over time in the relative number of CFU-GEMM cells, while these cells increased in CBA mice (Figure 3C). However, after immunization, a comparable change in the number of CFU-GEMM cells was observed for CBA and Th mice (Figure 3D). The relative number of CFU-GM cells in CBA mice at time zero was approximately 1.3- and 1.7-fold higher than in C57BL/6 and Th mice, respectively (Figure 3A). However, after immunization, a gradual decrease of CFU-GM cells occurred in CBA mice over time and a sharp decrease was observed in C57BL/6 mice over a period of 7–14 days, while a sharp increase of CFU-GM cells was observed in Th mice over the same period (Figure 3B). Thus, the change in the differentiation profile of bone marrow stem cells occurred in different ways for the three lines of mice before and after immunization. C57BL/6, CBA, and Th mice showed only some similarities for various precursors of hemopoietic cells. Nevertheless, these changes often followed a similar pattern in CBA and Th mice. In addition, and in contrast to CBA mice (1.05-fold), immunizing C57BL/6 mice (2.61-fold) and Th mice (1.59-fold) led to a growth in the total number of the four blood precursors in the brain (Figure 4B,C). Therefore, we cannot exclude that the increase in the number of precursors in the bone marrow of C57BL/6 and Th mice plays an important role in the development of autoimmune reactions.

Interestingly, the spontaneous development of autoimmune reactions in C57BL/6 mice is associated with a 6.5-fold increase in the number of CFU-E cells by day 7. Over the same period, the number of CFU-E cells decreased in CBA and Th mice before immunization, while a very slight change was observed after immunization (Figure 3C,D). The cell differentiation profile changes that are visible in the cerebrospinal fluid of patients with MS are associated with an increase of lymphocytes producing not only auto-Abs without activity but also abzymes [27,28,29]. IgGs from the liquor of MS patients hydrolyzing MBP, DNA, and polysaccharides are on average from 40 to 60 times more active than those taken from the sera of the same patients. We cannot exclude that during spontaneous and MOG-induced development of EAE in C57BL/6 mice the strong change in the number of brain CFU-E cells over time affects the production of specific lymphocytes synthesizing harmful abzymes hydrolyzing DNA, MBP, and MOG in mouse bone marrow.

Various cells from the bone marrow, including lymphocytes, enter the bloodstream and various organs of mammals. Therefore, we set out to compare the level of lymphocyte proliferation in the bone marrow and other organs of C57BL/6, CBA, and Th mice.

Notably, the total number of T and B lymphocytes before and after immunization increased significantly only in EAE-prone C57BL/6 mice (Figure 5A,B). Before immunization, the percentage content of lymphocytes changed slightly in the brain of Th mice, while it significantly decreased in CBA mice (Figure 5A). MOG treatment of Th and CBA mice led to a comparable decrease in the total number of brain lymphocytes (Figure 5B). Consequently, only in EAE-prone C57BL/6 mice the reorganization of the differentiation profile of brain stem cells was coupled with an increased proliferation of brain lymphocytes. The production of harmful abzymes with high catalytic activities may occur only in the brain of C57BL/6 mice, similar as seen in the liquor of MS patients [27,28,29].

Although the primary differentiation of lymphocytes occurs in the cerebrospinal fluid, their additional differentiation takes place in the blood and various organs of mammals. Therefore, we evaluated the level of lymphocyte proliferation in several organs. Spontaneous development of EAE in C57BL/6 mice led to an increase in the total number of lymphocytes in the spleen (1.8-fold; Figure 5C) and thymus (1.3-fold; Figure 6A), while their content decreased in the lymph nodes (1.8-fold) (Figure 6C). The changes in the number of spleen lymphocytes in CBA and Th mice before and after immunization did not match but were to some extent similar (Figure 5C,D). In contrast to C57BL/6 mice, lymphocytes in CBA and Th mouse spleens responded only weakly to MOG immunization.

Interestingly, the levels of thymus lymphocytes in CBA mice did not remarkably differ before and after MOG treatment (Figure 6A,B). This was similar for lymph node lymphocytes in Th mice (Figure 6C,D). At the same time, there was a lymphocyte increase in CBA mouse lymph nodes after immunization between days 7–14. Thus, the significant increase in the number of lymphocytes after immunization occurred in all organs only for EAE-prone C57BL/6 mice. The effect of MOG immunization on lymphocyte proliferation in all organs of Th and CBA mice was much weaker than the effects seen in C57BL/6 mice. We assume that the formation of specific lymphocytes predisposed to enhanced proliferation under the influence of MOG in different organs already occurs at the level of bone marrow differentiation in C57BL/6 mice, where the level of cell proliferation is increased significantly.

Time-dependent changes in the average relative activities of IgGs from the blood of C57BL/6, CBA, and Th mice were compared. Before immunizing Th mice, the change over time for DNase activity of IgGs was very low (Figure 9A-a1), similar to the change for anti-DNA Abs (7A-a2). After treating Th mice, the activity significantly increased between day 7 and days 20–25, comparable with changes seen in C57BL/6 mice in the acute EAE phase (Figure 9A-a1 and -a2). Interestingly, strong changes in DNase activity during this period in Th and C57BL/6 mice (Figure 9) were associated with relatively small changes in anti-DNA antibody concentrations (Figure 7). The subsequent increase in the relative concentration of anti-DNA Abs (>20–30 days) in the blood of C57BL/6 and Th mice (Figure 7A) led to a decrease in DNase activity of Abs (Figure 9A-a1). This means that during the period corresponding to the initial and acute phases of the EAE in C57BL/6 mice mainly abzymes appeared in the blood of mice, while in the remission period mainly auto-Abs without DNase activity appeared. In CBA mice, the situation was somewhat different; the relative concentration of anti-DNA Abs increased mainly up to day 14 and then reached a plateau (Figure 7A-a2). At the same time, the DNase activity of CBA mice Abs gradually increased up to day 60 (Figure 9A-a2). Thus, after immunizing CBA mice the formation of anti-DNA Abs without catalytic activity occurred first, followed by a slow rise in the concentration of abzymes hydrolyzing DNA in the blood of mice over 60 days. This is an important difference between C57BL/6 mice and Th and CBA mice in the synthesis of abzymes with DNase activity. The similarity between C57BL/6 mice and Th mice in the production of abzymes with DNase activity between days 7–20 may indicate that Th mice are more predisposed to develop autoimmune reactions leading to the production of DNase abzymes, similar as seen for C57BL/6 mice, and unlike CBA mice.

Notably, the level of DNA-, MBP-, and MOG-hydrolyzing activities in C57BL/6 mice gradually increased before immunization and during the spontaneous development of EAE (Figure 9). In contrast to C57BL/6 mice, the titers of Abs against DNA, MBP, and MOG in CBA and Th mice (Figure 7), as well as the activity of abzymes hydrolyzing these substrates (Figure 9), changed relatively slightly over time before immunization. Immunizing Th and CBA mice led to a remarkable increase of the Abs titers against these antigens up to approximately day 14, followed by a plateau (Figure 7). After immunizing C57BL/6 mice, the onset and acute phase of EAE revealed a sharp increase in Abs titers against both MBP and MOG (Figure 7B,C) and, especially, in the activity of IgGs hydrolyzing these substrates (9B,C). In contrast, immunizing CBA mice led to a consistent increase in the concentration of Abs against MBP and MOG between days 7–20, reaching a plateau thereafter (Figure 7, Figure 9). In contrast to CBA mice, a gradual increase in MOG-hydrolyzing activity was observed for Th mice up to day 60 (Figure 9B-b2), while Abs titers against MOG weakly increased in this time (Figure 7B-b2). Thus, in the initial period after immunization, mainly Abs without catalytic activity were formed, while the number of abzymes hydrolyzing MOG increased slowly during 80 days (Figure 7B and Figure 9B).

As described above, Th mice showed some similarity with C57BL/6 mice, demonstrating a sharp increase in DNase activity during the onset and acute phase of AI reactions. In contrast to CBA mice, IgGs of Th mice revealed MBP-hydrolyzing activity at time zero of the experiment. In addition, IgGs of Th mice, similar to C57BL/6 mice, exhibited a sharp increase in MBP-hydrolyzing activity during the onset and acute phase of AI reactions, followed by a decrease in the activity (Figure 9C-c2).

As shown above, all three mouse strains demonstrated different bone marrow stem cell differentiation profiles before and after their immunization with MOG (Figure 2 and Figure 3). Differentiation profiles for Th and CBA mice were more similar to each other than those for C57BL/6 mice. The most striking difference in the differentiation profiles of C57BL/6 mice compared to Th and CBA mice were the constant growth of CFU-E cells during their spontaneous and very sharp increase after immunization (Figure 2C,D). A significant difference between Th and CBA mice was the sharp increase in the number of BFU-E (Figure 2A,B) and GFU-GM cells (Figure 3A,B) after immunizing Th mice, while in CBA mice the change followed a different pattern. In addition, in CBA mice the number of CFU-GEMM cells increased before immunization, while in Th mice it decreased (Figure 3C). As shown above, in contrast to CBA mice (1.05-fold), immunization of C57BL/6 mice (2.61-fold) and Th mice (1.59-fold) led to an increase in the total number of the four blood precursors (Figure 4B,C). Additionally, a formation of different lymphocytes occurred in the liquor of various mice. Currently, it is not easy presently understood how the above-described differences in stem cell differentiation result in the formation of lymphocytes with different properties in the bone marrow and blood of C57BL/6, CBA, and Th mice.

It could be assumed that common to all autoimmune diseases in humans and experimental animals is that autoimmune reactions begin after changes in the differentiation profiles of bone marrow stem cells associated with the production of abzymes harmful to humans. It should be noted that the changes in the HSCs differentiation profiles during the spontaneous development of SLE in MRL-lpr-lpr and EAE in C57BL/6 mice, as well as after their immunization with DNA and MOG, respectively, are very similar [41]. There is some similarity in changing the HSCs differentiation profile in EAE-prone C57BL/6 and Th mice. Moreover, a common feature for SLE and MS patients and experimental animals prone to the development of these AIDs is the spontaneous production of abzymes that are dangerous since they stimulate the development of these diseases [8,9,10,11]. Abzymes with DNase activity in SLE and MS patients are cytotoxic, can penetrate the cell nucleus and hydrolyze DNA [7,11]. This leads to cell death due to cell apoptosis, which, in turn, stimulates the development of AIDs [30,31,32]. Abzymes with MBP- hydrolyzing activity are also noxious, because they can hydrolyze this protein of the axon’s myelin-proteolipid sheath which disrupts axonal nerve impulse conduction. Only at first glance, some AIDs are completely different, including SLE and MS. It should be mentioned that neuropsychiatric involvement occurs in MS, but also in about 50% of SLE patients and carries a poor prognosis (reviewed in [1]). Several indicators of disease common to SLE and MS were observed [1]. In addition, DNA- and MBP-hydrolyzing abzymes were found in SLE and MS patients [24,25]. Moreover, abzymes hydrolyzing DNA and MBP was revealed in sera of patients with schizophrenia also demonstrating neuropsychiatric components [44,45]. Thus, changes in the HSCs differentiation profiles and the intensity of production of abzymes with various activities in patients with different AIDs and experimental animals prone to the development of autoimmune reactions may vary in one way or another. However, most likely that in all cases, the development of these pathologies begins with a change in the differentiation profile of HSCs associated with the production of harmful abzymes.

## 5. Conclusions

Our data show that Th mice are less sensitive to the immunization with MOG and the development of autoimmune reactions than EAE-prone C57BL/6 mice but more predisposed to such reactions than CBA mice. It is very likely that for the development of autoimmune diseases, mammalian cells with a T- and B-cell response are required. In the future, we plan to carry out a more detailed analysis of the mechanisms of EAE development by comparing the various parameters described above for Th mice with T cell answer, for a line of mice with B cell response, and for a hybrid mouse strain obtained by crossing these two lines.

## Figures and Tables

**Figure 1 biomolecules-10-00053-f001:**
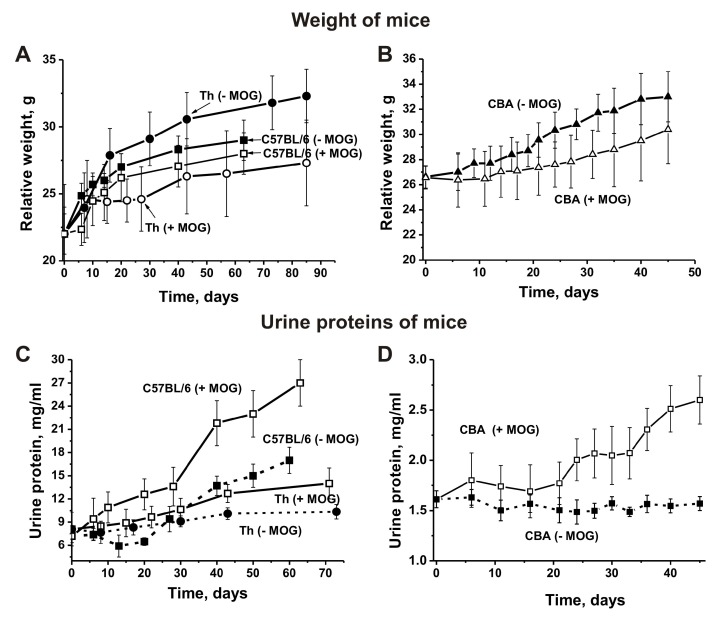
Relative changes in body weight over time characterize EAE-prone C57BL/6, Th (**A**), and CBA (**B**) mice before and after their immunization with myelin oligodendrocyte glycoprotein (MOG). Additionally, changes in proteinuria over time characterize EAE-prone C57BL/6, Th (**C**), and CBA (**D**) mice before and after their immunization with MOG.

**Figure 2 biomolecules-10-00053-f002:**
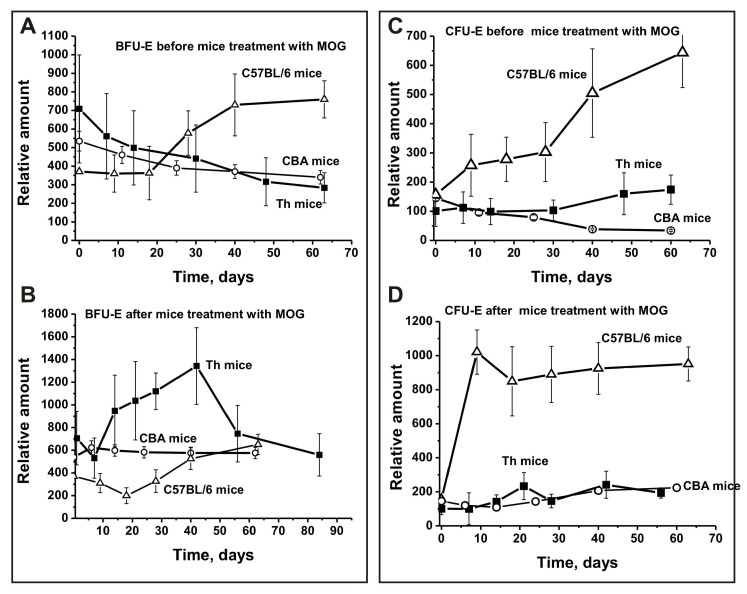
Changes over time in the number of mouse brain BFU-E (**A**,**B**) and CFU-E (**C**,**D**) cells before and after MOG treatment of C57BL/6, CBA, and Th mice. BFU-E: erythroid burst-forming unit, early erythroid colonies; CFU-E: erythroid burst-forming unit, late erythroid colonies.

**Figure 3 biomolecules-10-00053-f003:**
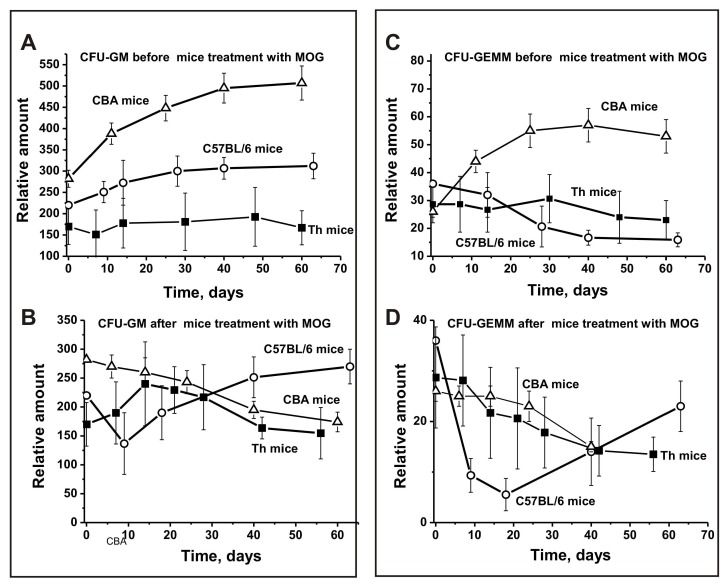
Changes over time in the number of mouse brain CFU-GM (**A**,**B**) and CFU-GEMM (**C**,**D**) cells before and after MOG treatment of C57BL/6, CBA, and Th mice. CFU-GM: granulocytic-macrophagic colony-forming unit; CFU-GEMM: granulocytic, erythroid, myeloid colony-forming unit.

**Figure 4 biomolecules-10-00053-f004:**
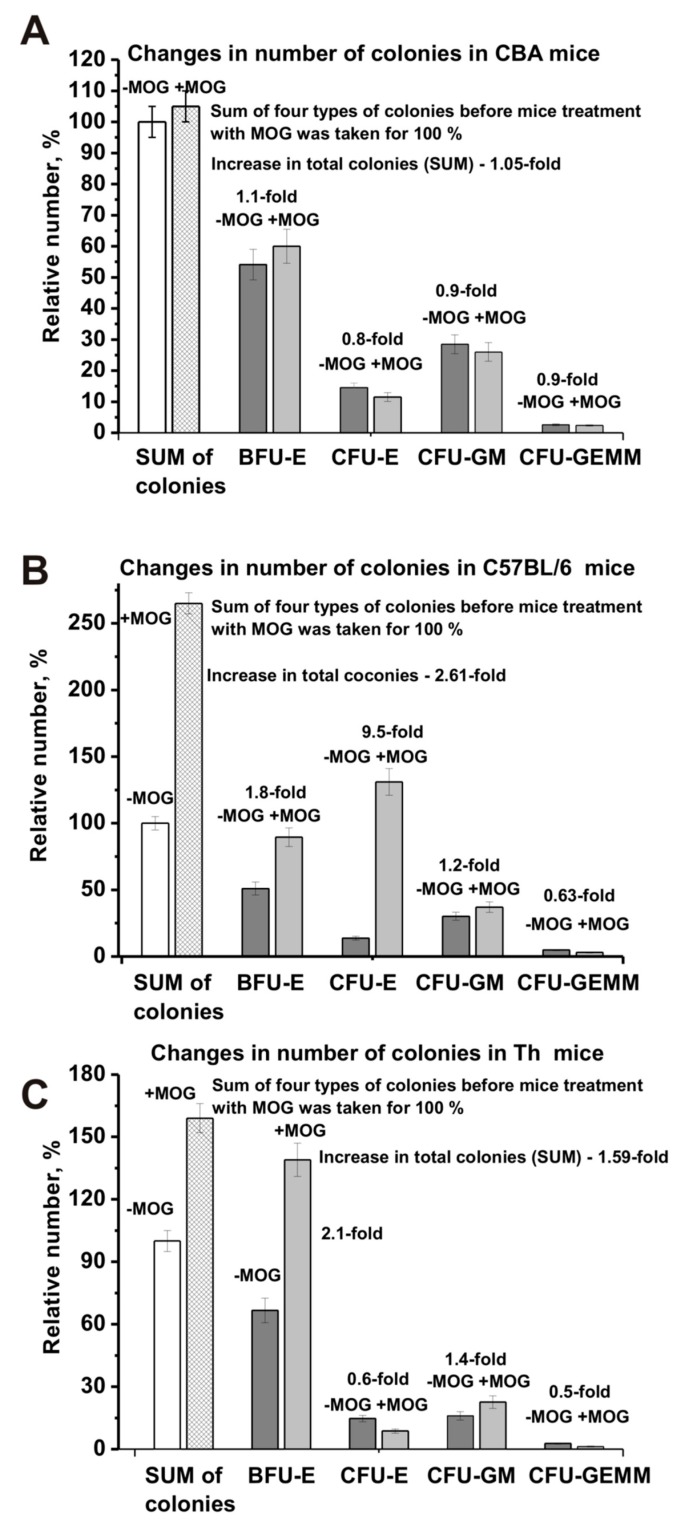
Changes in the sum of the number of BFU-E, CFU-E, CFU-GM, and CFU-GEMM cells before and after MOG treatment: CBA (**A**), EAE-prone C57BL/6 (**B**), and Th (**C**) mice. The sum of all four types of colonies before immunization was taken as 100%.

**Figure 5 biomolecules-10-00053-f005:**
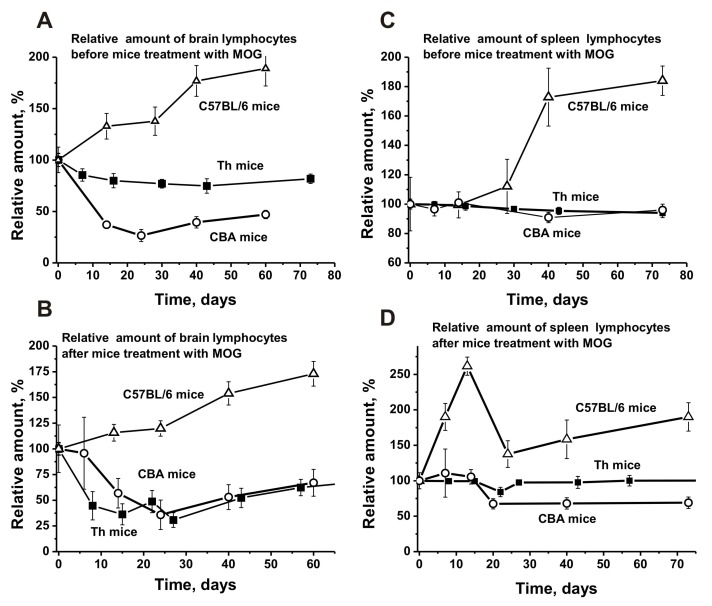
The average changes over time in the relative number of lymphocytes before and after treatment with MOG in the bone marrow (**A,B**) and spleen (**C,D**) of C57BL/6, CBA, and Th mice. First, the number of lymphocytes was estimated from optical density for each mouse and all groups (with seven mice per group); the error from three independent experiments did not exceed 7–10%. Then the relative number of lymphocytes (optical density) before immunization at time zero of the experiment was taken as 100%.

**Figure 6 biomolecules-10-00053-f006:**
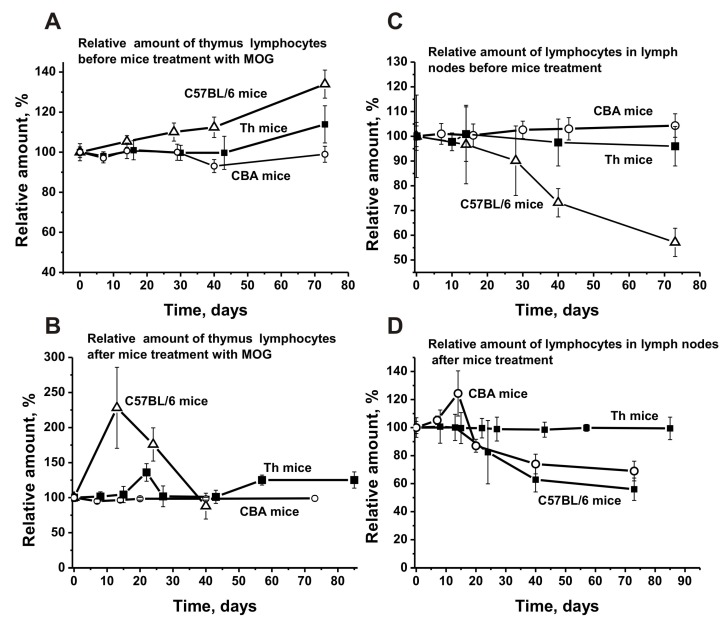
The average changes over time in the relative number of lymphocytes before and after treatment with MOG in the thymus (**A,B**) and lymph nodes (**C,D**) of C57BL/6, CBA, and Th mice. First, the relative number of lymphocytes was estimated from optical density for each mouse and all groups (with seven mice per group); the error from three independent experiments did not exceed 7–10%. Then the relative number of lymphocytes (optical density) before immunization at time zero of the experiment was taken as 100%.

**Figure 7 biomolecules-10-00053-f007:**
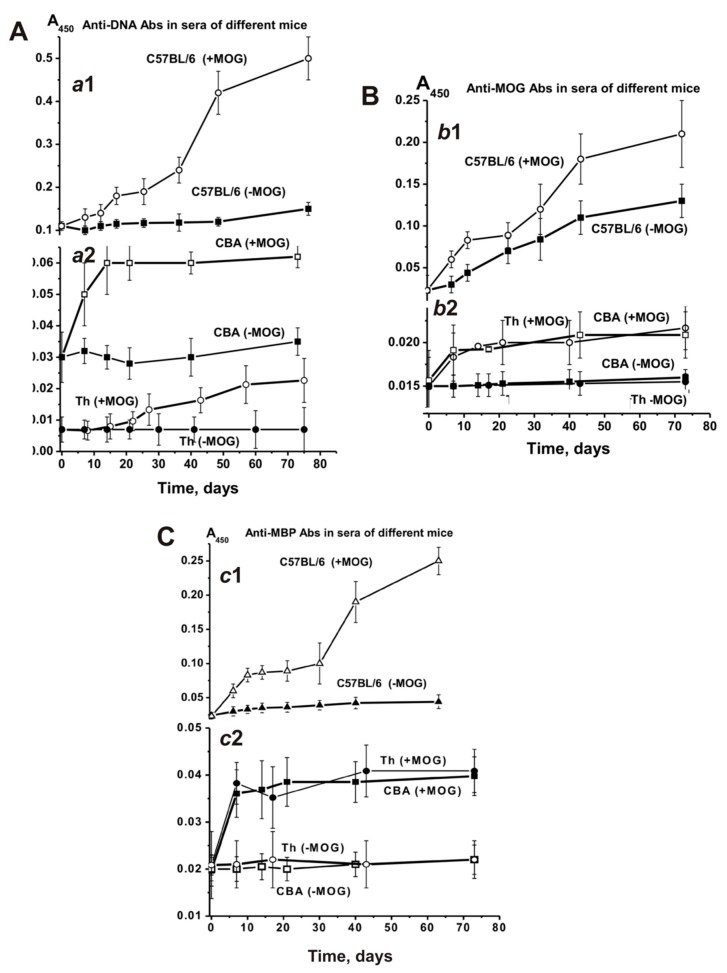
Relative changes over time in anti-DNA (**A**; a1 and a2), anti-MOG (**B**; b1 and b2), and anti-MBP (**C**; c1 and c2) concentrations characterizing EAE-prone C57BL/6, CBA, and Th mice before and after their immunization with MOG. The relative concentration was estimated for each mouse and all groups (with seven mice per group); the error from three independent experiments did not exceed 7–10%.

**Figure 8 biomolecules-10-00053-f008:**
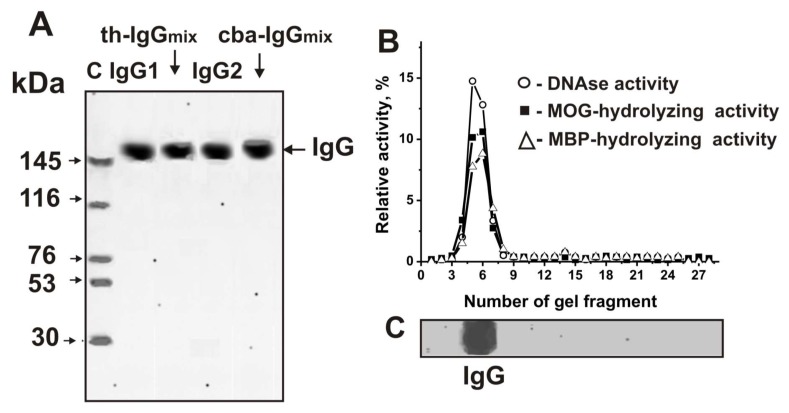
The homogeneity analysis by SDS-PAGE of 10 μg IgG1 and IgGmix corresponding to Th mice and 10 μg IgG1 and IgGmix corresponding to CBA mice under non-reducing conditions (**A**); silver staining. The arrows (lane C (A)) show the positions of proteins with known molecular masses. The relative activities (%) of eluate gel fragments in the hydrolysis of DNA (o), MOG (■), and MBP (∆) by th-IgGmix were determined using the extracts of gel fragments (2–3 mm) (**B**). The position of IgGs is shown on Panel, **C**. A complete hydrolysis of all substrates after incubation was taken as 100% (B). The errors of the relative activity determined from two independent experiments did not exceed 7–10%.

**Figure 9 biomolecules-10-00053-f009:**
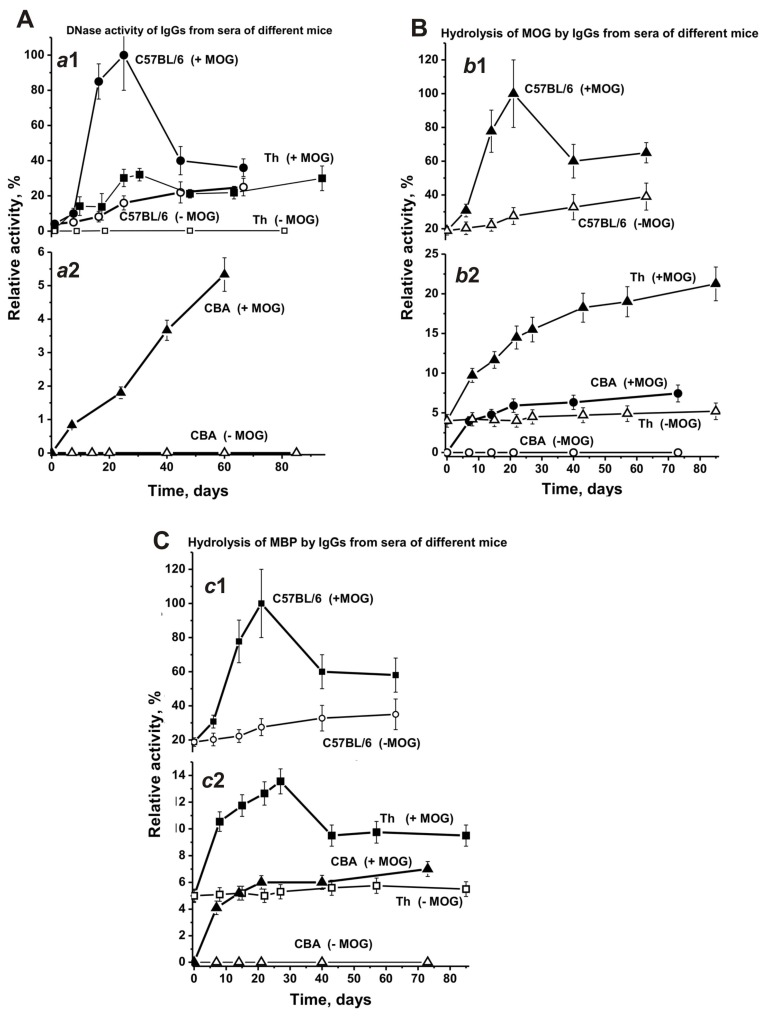
Changes over time in the average relative activities of IgGs hydrolyzing DNA (**A**; a1 and a2), MOG- (**B**; b1 and b2), and MBP (**C**; c1 and c2) for Abs from C57BL/6, CBA, and Th mouse groups (each group made up of 7 mice) before and after their treatment with MOG. The error in the initial rates determined from two experiments for each mouse for all groups did not exceed 7–10%.

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
