# Peer review of "Production of Abzymes in Th, CBA, and C57BL/6 Mice before and after MOG Treatment: Comparing Changes in Cell Differentiation and Proliferation"

_biomolecules, 2019, doi:10.3390/biom10010053_

Round 1

Reviewer 1 Report

This is an interesting paper that investigates the production of abzymes in different mice populations before and after MOG treatment.

I only had a few questions that I would like the authors to comment on:

It wasn`t clear from the paper which group of the mice acted as the control group, could this be emphasized in the manuscript.  Since MOG was used for immunisation shouldn`t the study be more centred towards neurological autoimmune diseases?  It wasn`t clear why only C57BL/6  mice were predisposed to MOG induced EAE development and why the other mice species were less susceptible- more explanation warranted.  Why did the authors choose the 3 types of mice in the study and not other mice?  How will the findings of this study impact on autoimmune diseases in humans? This wasn`t very clear from the discussion?  Some information should be provided on the interaction of T and B lymphocytes and the development of autoimmune disease.  What mechanisms of EAE development were identified in the present study?  The abstract should be rewritten with a clearer introduction and induce some actual results from the study.

Author Response

Dear Editor,

Thank you for considering our manuscript for publication. We have tried to take into account all remarks of the reviewers. We are now sending the corrected version with the changes outlined below.

First referee

I only had a few questions that I would like the authors to comment on:

It wasn`t clear from the paper which group of the mice acted as the control group, could this be emphasized in the manuscript.  Since MOG was used for immunisation shouldn`t the study be more centred towards neurological autoimmune diseases?  It wasn`t clear why only C57BL/6  mice were predisposed to MOG induced EAE development and why the other mice species were less susceptible- more explanation warranted.  Why did the authors choose the 3 types of mice in the study and not other mice?  How will the findings of this study impact on autoimmune diseases in humans? This wasn`t very clear from the discussion?  Some information should be provided on the interaction of T and B lymphocytes and the development of autoimmune disease.  What mechanisms of EAE development were identified in the present study?  The abstract should be rewritten with a clearer introduction and induce some actual results from the study. 

Answer:

As shown earlier in [35] and in several our works [7-10, 14-18], the development of autoimmune diseases is associated with defects of the immune system of the bone marrow, namely, a change in the profile of differentiation of bone marrow stem cells. These data were given on page 3.

It is well-established that all autoimmune pathologies arise after a self-tolerance breakdown (central or peripheral) and progress via multiple mechanisms. Also, autoimmune diseases have been suggested to originate from hematopoietic stem cell (HSC) defects [35]. It has been shown that the spontaneous and antigen-induced development of profound EAE pathology in C57BL/6 mice [17,18] and SLE pathology in MRL-lpr-lpr mice [14-16] is coupled with an immune system-specific reorganization. This reorganization includes irregular changes in the differentiation profile of bone marrow HSCs in combination with the production of abzymes hydrolyzing DNA, ATP, polysaccharides, and proteins. This phenomenon is one of the multiple mechanisms underlying self-tolerance breakdown.

It should be noted that we have two groups of mice with different cellular responses: Th with a T-cell response and C57Bl/6JRj strain with a B-cell response. According to scientific groups in Germany, in general terms, the development of EAE in these two lines of mice is very slow. However, the crossbreeding of Th and C57Bl/6JRj strain mice leads to hybrids in which EAE develops rapidly. We plan to analyze parameters characterizing the development of autoimmune reactions in Th and C57Bl/6JRj  mice, as well as their hybrids. In this article we have analyzed only Th mice with a T-cell response. Characterization of Th mice was given on page 4.

Here we used a model of spontaneous CNS autoimmunity which was generated by crossing myelin-specific T-cell receptor (TCR) transgenic mice and myelin-specific Ig heavy chain knock-in mice [39]. The offspring of those mice (Th mice) are characterized by spontaneous development of a severe form of EAE.

Taking into account your remark we have added the following text on page 4.

Additional Information: As shown in [14-16], in  the study of SLE-prone MRL-lpr-lpr mice, in control CBA mice, in contrast to MRL-lpr-lpr mice, autoimmune reactions and significant changes in the differentiation profile after their immunization with DNA do not occur. At the same time, it was shown in [17-18] that in C57BL/6 mice with T- and B-cells response predisposed to EAE, the development of pathology is associated with a change in the profile of differentiation of bone marrow stem cells and the production of abzymes. Th mice with T-cells response also predisposed to spontaneous development of EAE [39]. Therefore, mice predisposed not predisposed to autoimmune to AIDS were selected as two control groups.

From our point of view, this choice turned out to be justified, because Th mice showed parameters characterizing tested before and after immunization with MOG intermediate between CBA and C57BL/6 mice. Some of the indicators were close to those for CBA, and the other for C57BL/6 mice.

As for the neurological indicators of the development of EAE, two groups in Germany (Thomas Budde and Sven G. Meuth) deal with this aspect and these data partially published [39] and some of them will be published later. In this work, it is too early to talk about the interaction of T and B lymphocytes in the development of an autoimmune disease. We are currently engaged in the analysis of T and B lymphocytes of Th mice before and after their immunization with MOG and it will be published elsewhere. However, a clearer answer about the interaction and role of T- and B-lymphocytes, as well as the production of abzymes harmful to mice, can we get only after studying Th and C57Bl/6JRj strain mice and their hybrids. It is necessary to understand the difference in the lines with the T- and B-cell response and their hybrid, and then everything will come together.

Sincerely Prof. Georgy A. Nevinsky

Reviewer 2 Report

The paper of K. Aulova et al. describes effects of immunization of three different mouse strains with myelin oligodendrocyte glycoprotein (MOG). The authors analyzed hematopoetic stem cells and progenitors, cell numbers of lymphocytes in various organs, and concentrations of catalytic antibodies that hydrolyze DNA, MOG or MBP before and after immunization. The experiments are sound and described in great detail.

The authors observed differences between three different mouse strains. This is not surprising because it is known for long that different mouse strains have very different susceptibilities for EAE. However, although the lengthy manuscript describes many experiments, this manuscript is a collection of individual observations and it is not clear why some of the parameters were chosen while others were not. Further, it is not clear how hematopoetic progenitors and catalytic antibodies are functionally linked. Taken together, the authors could not find a common denominator that might reveal a basic pattern on the production of abzymes, let alone on susceptibility or pathogenesis of EAE. The authors should try to address these points in the Introduction and Discussion.

Author Response

Dear Editor,

Thank you for considering our manuscript for publication. We have tried to take into account all remarks of the reviewers. We are now sending the corrected version with the changes outlined below.

Second referee

The paper of K. Aulova et al. describes effects of immunization of three different mouse strains with myelin oligodendrocyte glycoprotein (MOG). The authors analyzed hematopoetic stem cells and progenitors, cell numbers of lymphocytes in various organs, and concentrations of catalytic antibodies that hydrolyze DNA, MOG or MBP before and after immunization. The experiments are sound and described in great detail.

The authors observed differences between three different mouse strains. This is not surprising because it is known for long that different mouse strains have very different susceptibilities for EAE. However, although the lengthy manuscript describes many experiments, this manuscript is a collection of individual observations and it is not clear why some of the parameters were chosen while others were not. Further, it is not clear how hematopoetic progenitors and catalytic antibodies are functionally linked. Taken together, the authors could not find a common denominator that might reveal a basic pattern on the production of abzymes, let alone on susceptibility or pathogenesis of EAE. The authors should try to address these points in the Introduction and Discussion.

Answer:

At this stage, it is difficult to answer to some questions. However, it should be noted that, as shown earlier in [35] and by us in several works [7-10, 14-18], the development of autoimmune diseases is associated with a violation of the immune system of the bone marrow, namely, a change in the differentiation profile of bone marrow stem cells. These data were given on page 3.

It is well-established that all autoimmune pathologies arise after a self-tolerance breakdown (central or peripheral) and progress via multiple mechanisms. Also, autoimmune diseases have been suggested to originate from hematopoietic stem cell (HSC) defects [35]. It has been shown that the spontaneous and antigen-induced development of profound EAE pathology in C57BL/6 mice [17,18] and SLE pathology in MRL-lpr-lpr mice [14-16] is coupled with an immune system-specific reorganization. This reorganization includes irregular changes in the differentiation profile of bone marrow HSCs in combination with the production of abzymes hydrolyzing DNA, ATP, polysaccharides, and proteins. This phenomenon is one of the multiple mechanisms underlying self-tolerance breakdown.

It should be noted that we have two groups of mice with different cellular responses: Th with a T-cell response and C57Bl/6JRj  with a B-cell response. According to scientific groups in Germany, in general terms, the development of EAE in these mice is very slow. However, the crossbreeding of Th and C57Bl/6JRj  mice leads to hybrids in which EAE develops rapidly. We plan to analyze as parameters characterizing the development of autoimmune reactions in Th and C57Bl/6JRj  mice, as well as their hybrids. This paper analyzes only Th mice. As shown in our works [14-16] on the study of SLE-prone MRL-lpr-lpr mice, in control CBA mice, in contrast to MRL-lpr-lpr mice, autoimmune reactions and significant changes in the differentiation profile after their immunization with DNA do not occur. At the same time, it was shown in [17-18] that, in mice predisposed to EAE C57BL/6 mice, the development of pathology is associated with a change in the profile of differentiation of bone marrow stem cells and the production of abzymes. Considering this, mice that were not predisposed to autoimmune diseases and mice that were not predisposed to autoimmune diseases were selected as control groups. From our point of view, this choice turned out to be justified, because Th mice showed parameters characterizing the parameters we tested before and after immunization with MOG intermediate between CBA and C57BL/6 mice. Some of the indicators were close to those for CBA, and the other for C57BL/6 mice.

As we have shown earlier,  that the appearance of abzymes is a statistically reliable indicator of the onset and development of various autoimmune diseases and their production is associated with a change in the differentiation profile of bone marrow stem cells. Given this, in this paper we focused on the analysis of these indicators. As for the neurological indicators of the development of EAE, two groups in Germany (Thomas Budde and Sven G. Meuth) deal with this aspect and these data partially published [39] and some of them will be published later. In this work, it is too early to talk about the interaction of T and B lymphocytes and the development of an autoimmune disease. We are currently engaged in the analysis of T and B lymphocytes of Th mice before and after their immunization with MOG. However, a clearer answer about the interaction and role, as well as the production of abzymes harmful to mice, can we get only after studying C57Bl/6JRj  and their hybrids with Th mice. A common denominator indicating the basic patterns of abzyme production in the pathogenesis of autoimmune diseases is a change in the bone marrow stem cell differentiation profile. In work [41], we showed that the change in the profile of stem cells differentiation during spontaneous development of SLE in MRL-lpr-lpr mice after treatment with DNA, as well as after immunization C57BL/6 mice with MOG, is the same despite the fact that the beginning of SLE is associated with the production of antibodies and abzymes against DNA, and in C57BL/6 mice, both against the myelin basic protein and DNA. More detailed information can be obtained only after comparing Th and C57Bl/6JRj mice and their hybrids.

Hematopoetic progenitors and lymphocytes of brain from our point of view are functionally linked with catalytic antibodies. Changes in the differentiation profile of brain stem cells leads to the production harmful to humans catalytic antibodies, hydrolyzing MBP and DNA, not only in the blood, but already in the cerebrospinal fluid. Moreover, the activity of abzymes in the cerebrospinal fluid of patients with multiple sclerosis is 40-60 times higher than in the blood of the same patients. This information was given on page 3.

Levels of IgGs from the CSF of MS patients hydrolyzing MBP, DNA, and polysaccharides are on average ~40–60 times higher than those taken from the sera of the same patients [27-29].

Abzymes hydrolyzing DNA penetrate the nuclei of cells, initiate their apoptosis and, as a result, increase the concentration of DNA in the blood and anti-DNA abzymes. This leads to kindling of autoimmune disease. Abzymes against MBP hydrolyze this protein in the membranes of nerve tissues, which leads to impaired impulses along them. This information was given on page 3.

Taking into account your comments we have added additional information at the end of discussion? But we cannot add it to Summary, since there may be only 200 words

It could be assumed that common to all autoimmune diseases in humans and experimental animals is that they begin after changes in the differentiation profiles of bone marrow stem cells associated with the production of abzymes harmful to humans. It should be noted that the change in the HSCs differentiation profiles during the spontaneous development of SLE in a MRL-lpr-lpr and EAE in C57BL/6 mice, as well as after their immunization with DNA and MOG, respectively, are very similar [41]. There is some similarity in changing the HSCs differentiation profile in EAE-prone C57BL/6 and Тh mice. Moreover, a common feature for SLE and MS patients and experimental animals prone to the development of these AIDs is the spontaneous production of abzymes that are dangerous since they stimulate the development of these diseases [8-11]. Abzymes with DNase activity in SLE and MS patients are cytotoxic, can penetrate the cell nucleus and hydrolyze DNA [7,11]. This leads to cell death due to cell apoptosis, which, in turn, stimulates the development of AIDs [30-32]. Abzymes with MBP- hydrolyzing activity are also noxious, because they can hydrolyze this protein of the axon’s myelin-proteolipid sheath which disrupts axonal nerve impulse conduction. Only at first glance, some AIDs are completely different, including SLE and MS. It should be mentioned that neuropsychiatric involvement occurs in MS, but also in about 50% of SLE patients and carries a poor prognosis (reviewed in [1]). Several indicators of disease common to SLE and MS were observed [1]. In addition, DNA- and MBP-hydrolyzing abzymes were found in SLE and MS patients [24,25]. Moreover, abzymes hydrolyzing DNA and MBP was revealed in sera of patients with schizophrenia also demonstrating neuropsychiatric components [44,45]. Thus, changes in the HSCs differentiation profiles and the intensity of production of abzymes with various activities in patients with different AIDs and experimental animals prone to the development of autoimmune reactions may vary in one way or another. However, most likely that in all cases, the development of these pathologies begins with a change in the differentiation profile of HSCs associated with the production of harmful abzymes.

Sincerely Prof. Georgy A. Nevinsky

Round 2

Reviewer 1 Report

The authors have provided sufficient answers to my questions and have amended their manuscript accordingly and therefore making it appropriate for publication.

Reviewer 2 Report

The paper has improved and may pass now.